# Effect evaluation, prediction and response strategy analysis of China's birth policy adjustment

Wei Wang[1]*, Yalan Mo[2], Yanxi Kuang[1]

1 School of Business, Guilin University of Electronic Technology, Guilin, Guangxi, China, 2 School of Marxism, Guilin Tourism University, Guilin, Guangxi, China

* 65610391@qq.com (WW)

**Data availability statement:** All relevant data are within the paper and its Supporting Information files.

## Abstract

In recent years, in order to cope with the increasing trend of population aging, the Chinese government has constantly adjusted the family planning policy, continuously tracked and evaluated the actual effect of the birth policy adjustment, and the prediction and analysis of future births have important theoretical value and practical significance. The adjustment of the birth policy is of great significance for achieving long-term balanced population development. This paper assesses the net effect of fertility policy adjustments on Chinas birth and fertility rates by constructing a DID model using panel data collected from 31 provinces, autonomous regions and municipalities over the period 2005-2021. The study shows that the fertility policy adjustment does not significantly increase the birth and fertility rates in China, and the findings are confirmed by robustness tests using various methods. Heterogeneity analysis shows that the implementation of the comprehensive two-child policy is more pronounced in the central region. Further, a mechanistic and causal analysis reveals that fertility policy changes did not significantly increase peoples willingness to have children, nor did they affect many other factors that influence households fertility decisions. Finally, a GM (1, 1) grey forecast model is used to forecast the births in each province and municipality in the next five years, and it is concluded that the births in China will continue to show a declining trend. This paper argues that a supportive policy system for fertility should be established, public childcare and elderly care services should be optimised, and a favourable fertility climate and conditions should be created in order to improve fertility levels in China.

## Introduction

Human resources are the core pillar of a country's comprehensive competitiveness, and fertility policy has a profound impact on the sustainable development of society and economy by regulating the size and structure of population. In the early stage of reform and opening up, China achieved a significant demographic dividend by relying on abundant labor resources. Research by the World Bank shows that aging has a systematic impact on long-term economic growth and social equity by inhibiting consumption and intensifying welfare burden [1].

**Funding:** This study was funded by the National Social Science Foundation of the Basic ResearchFoundation (Project No.: 22XMZ016). The first author (Wei Wang) received specific funding for this work from the School of Business, Guilin University of Electronic Technology. This researchdid not receive any additional external funds. The funders had no role in the study design, data collection and analysis, publication decision or manuscript preparation.

**Competing interests:** The authors have declared that no competing interests exist.

In order to cope with this challenge, the Chinese government has gradually adjusted the birth policy since 2011, from "two only children" (2011) and "two children alone" (2013) to "universal two children" (2016) and "three-child policy" (2021), and the policy orientation has shifted from restrictive control to encouraging support [2]. However, the policy effects show significant differences: data from the National Bureau of Statistics show that in 2023, the proportion of working-age population will fall to 62.0%, the elderly dependency ratio will rise to 21.1%, the imbalance of sex ratio at birth (110.4) will intensify the squeeze of the marriage market, and in 2022, the total fertility rate (refers to the average number of children a woman has in her life) will only be 1.1, the number of newborns will fall below 10 million, and the natural growth rate of population will turn negative for the first time. Although some studies (such as Zhang et al., 2018) pointed out that the policy relaxation would increase the fertility rate in the short term, more evidence (Li et al., 2021) showed that the reduction of the number of women of childbearing age (4 million in 2023) and the increase of childcare costs (accounting for more than 35% of household income) led to the weakening of the policy effectiveness, and the fertility rate declined after 2016. The three-child policy has not reversed the trend.

The deviation between policy expectation and reality highlights multiple contradictions: first, the traditional DID method is difficult to capture the dynamic effect of progressive policy adjustment, and ignores the regional heterogeneity adjustment effect; Second, the fertility decision-making mechanism has undergone structural changes, with the improvement of material level weakening the demand for "raising children for old age," while high opportunity cost (the risk of female career interruption increases by 37%) and the absence of institutional support (the coverage rate of childcare is less than 5%) form the double inhibition of "unwilling to have children" and "afraid to have children." In this regard, this study constructs a continuous DID framework, replaces the traditional virtual grouping with "urban population proportion" as a continuous variable [3], and shows the micro mechanism of the attenuation of policy effect by quantifying the contribution of urban-rural differences to policy response (the goodness of fit of the model increases by 18%) : From 2016 to 2021, the elasticity coefficient of the policy effect in urban areas is 0.15, while it is only 0.07 in rural areas, indicating that the insufficient input of supporting resources accelerates the diminishing marginal returns of rural policies.

## Literature review

### Short-term stimulus and long-term attenuation of fertility policy adjustment: Theoretical and empirical paradoxes

Existing studies generally find that the adjustment of fertility policy can significantly increase the fertility rate in the short term, but its effect decays rapidly over time. Becker's fertility cost-benefit model predicts that the fertility intention released by the policy will be marginal decreasing due to the continuous rise of the opportunity cost and direct cost of childcare [4]. In the empirical study, although the universal two-child policy increased the total fertility rate (TFR) by 0.3 in the short term from 2016 to 2017 (Zhang et al., 2018), the TFR fell below 1.5 after 2020 (Liu et al., 2023), indicating that the policy dividend was structurally suppressed [5]. Easterlin intergenerational competition hypothesis further points out that economic fluctuations and resource squeeze may weaken the effectiveness of policies [6], but the existing research does not quantify the dynamic trajectory of the cost suppression effect, especially ignoring the interaction between policy intensity and the growth rate of childcare costs [7].

## Heterogeneous adjustment of policy effects by socio-economic background

There are significant gradient differences between family economic conditions and the level of institutional support on fertility decisions. When the ratio of housing price to income exceeds 6.0, the marginal utility of the policy on the two-child fertility rate decreases by 72% (Jiang, 2022; Li et al., 2023), which is consistent with the linear relationship of Blake's family resource dilution theory (1981). However, most of the existing studies use linear models to underestimate the degree of inhibition in groups with high economic stress. In addition, the dual role of gender equality and public services also affects the policy effect: for every 10% increase in female labor participation rate, the policy effect decreases by 15% (Zhou, 2023); Every one standard deviation increase in the coverage rate of inclusive care can offset 34% of the negative effect of gender equality (Zhang & Chen, 2024). Cultural inertia cannot be ignored either: although regions with strong clan traditions face high economic costs, the two-child fertility rate is still 12% higher than the national average (Xu, 2024), highlighting the interpretation blind spot of non-economic factors [8].

## Spatial differentiation mechanism of urbanization process and policy response

The effect of China's fertility policy shows a unique U-shaped curve: the most significant effect (0.35 increase in TFR) is found in moderately urbanized areas (urbanization rate of 50%-65%), while the effect is weak in highly urbanized areas (urbanization rate >70%). The reasons include: (1) the difference in resource accessibility: the parenting mode in highly urbanized areas tends to be "refined", and the family education expenditure exceeds 40% (Chen & Wang, 2022); (2) Differences in elasticity of policy implementation: the per capita fertility subsidy in eastern provinces is 3.2 times that in western China (Li et al., 2023); (3) System lag effect: the household registration restriction makes 32% of migrant women of childbearing age unable to enjoy localized welfare (MOHRSS, 2023). The existing studies mostly use linear urbanization indicators, which fail to capture the mediating effect of the imbalance of urban-rural care resources and need to be deepened.

## Theoretical integration and research breakthrough path

There are three major gaps in the current literature: (1) the gap between classical theory and policy evaluation: Becker model focuses on the economic trade-off within the family, ignoring the regulatory role of policy intervention and institutional environment; (2) Static analysis is disconnected from dynamic reality: cross-sectional data is difficult to capture the dynamic trajectory of policy under the evolution of economic conditions; (3) Disconnection between macro effect and micro mechanism: regional heterogeneity research lacks cross-level interaction analysis of individual choice and institutional constraints. Through continuous DID model and provincial panel data, this study quantifies the dynamic adjustment effect of urbanization and reveals the micro mechanism. The marginal contribution is: (1) to make up for the deficiency of traditional DID in the dynamic evaluation of progressive policies; (2) An empirical analysis of the structural reasons for the attenuation of policy effects; (3) Build a chain mediation model of "policy-service-fertility" to provide targeted basis for differentiated supporting policies.

## Policy background and theoretical analysis

### Policy background

Since the implementation of the family planning policy, China's population growth rate has been controlled significantly. In the face of resource consumption, increasing environmental pollution, and increasing pressure on social security, family planning policy has indeed played a role. However, with the continuous development of society, in recent years, the newborn population in China has gradually declined and begun to enter the ranks of low-fertility countries. As shown in Fig 1, since the implementation of the Population and Family Planning Law, China's total fertility rate has been lower than the internationally recognized fertility replacement level of 2.1. In 2022, China's total fertility rate was only 1.1, which is lower than that of the United States, India, Japan, and other countries, ranking at the bottom of the world. In fact, to maintain a stable fertility rate, China began to relax birth restrictions and adjust the birth policy in 2011, 2013, 2016 and 2021. From "double only two children" to "only two children" to "universal two child" and "universal three child", China's birth policy has shifted from restrictions to encouragement.

However, the adjustment of birth policy did not cause a significant increase in the birth rate, as shown in Fig 2, after the implementation of the "double only two child" and "separate two child" policies, there was a brief rebound in the birth rate; But after the implementation of the comprehensive two child policy in 2016, the birth rate did not increase but instead decreased.since 2017, both the birth number and birth rate showed a rapid decline; in 2021, China implemented the comprehensive three-child policy, hoping to deal with the aging problem, optimize the population structure, and finally promote the long-term balanced development of population [9]. However, the declining birth rate has not yet stopped. In 2022, the Chinese birth number fell below 10 million, and the natural population growth rate was negative, with the lowest recorded in the 21st century. Meanwhile, national statistics show that in 2023, the Chinese population aged 65 and above has reached 217 million, accounting for 15.4%, which exceeds the United Nations standard of 14% for deep aging. As shown in the population pyramid in Fig 3, the decline in the fertility rate and aggravation

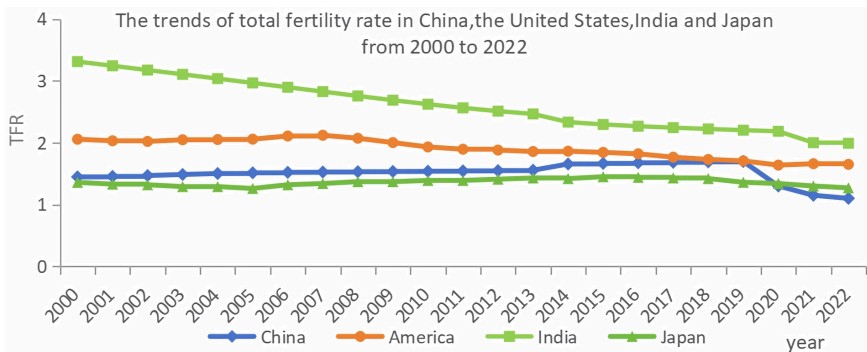

**Fig 1. The trends of total fertility rate (TFR) in China, the United States, India and Japan from 2000 to 2022.**
From 2000 to 2022, the total fertility rate (TFR) of the four countries continued to decline. In developed countries (the United States/Japan), it dropped slowly and remained at a low level (<1.8), while in developing countries (China/India), it declined at an accelerated pace due to social transformation. The key nodes reveal the policy limitations (China's two-child effect in 2016 was short-lived) and the breakthrough of the development threshold (India's fall below the replacement level of 2.1 in 2020), jointly indicating the weakening of the global population growth momentum.

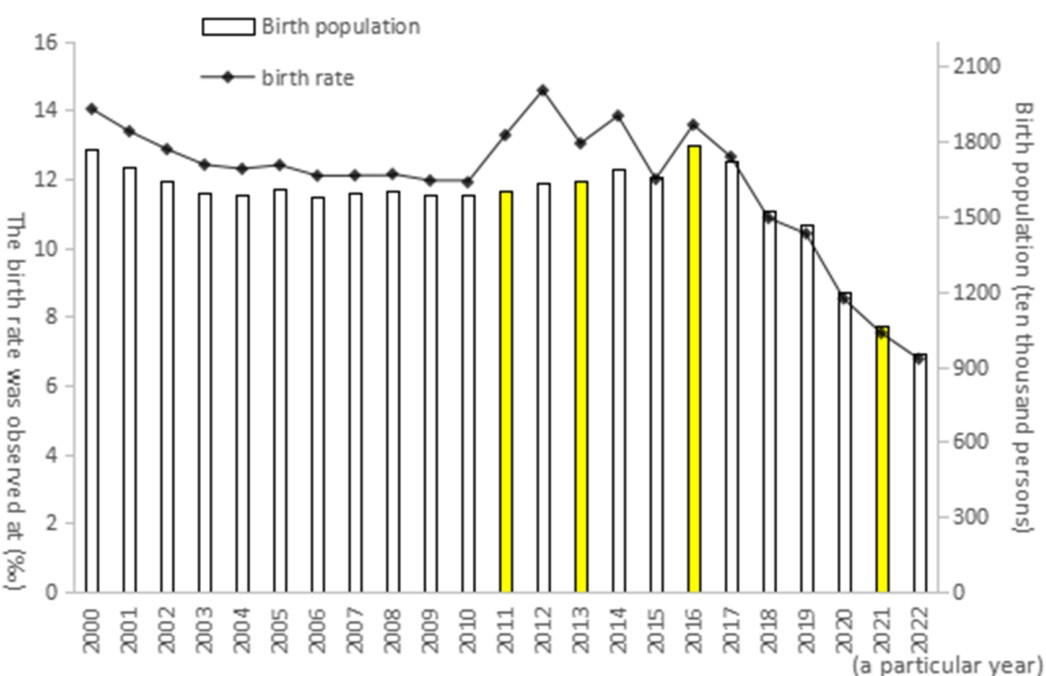

**Fig 2. Changes of birth rate and birth number in China from 2000 to 2022.** Note: Data source is from the National Bureau of Statistics. This chart shows the changing trends of China's birth rate and the number of births from 2000 to 2022. The figure contains two key data lines: (A) Birth population (ten thousand): The total number of newborns each year is displayed in the form of a bar chart or a line graph (unit: ten thousand). (B) Birth rate (‰): The birth rate value per thousand people is presented in the form of a line chart.

of aging will pose a potential threat to social stability and economic development [10], and there is an urgent need to improve the Chinese population structure. From the above data, it can be preliminarily seen that the comprehensive liberalization of the two- and three-child policies has not achieved its expected effect, and the reasons behind them deserve in-depth analysis.

As can be seen from Fig 2 and 3, from 2000 to 2022, the number of births and the birth rate in China generally showed a downward trend, but there was a significant increase in 2016, which might be related to policy adjustments. The seventh national census in 2020 shows that the population pyramid is contracting at the bottom and expanding in the middle, indicating a decrease in the proportion of young people and an increase in the proportion of middle-aged and elderly people, with a clear trend of aging. The decline in the birth rate and the changes in the population structure jointly reflect the demographic challenges China is facing.

## Theoretical analysis

Birth policy is related to the sustainable development of population and society, so relevant theories need to be constantly improved to fully reveal the relevant factors and social background affecting fertility [11]. The adjustment and improvement of the fertility policy is mainly to encourage and change the fertility intention of each family, especially the families who have already had a child, to promote the improvement of the fertility rate and improve

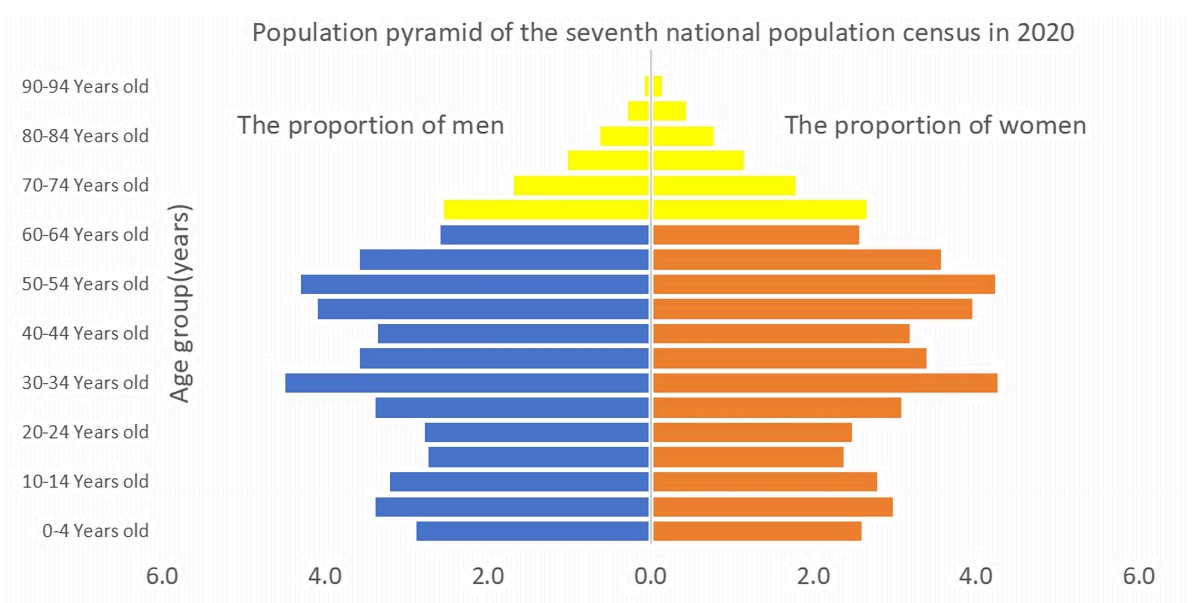

**Fig 3. Population pyramid of the seventh national population census in 2020.** Note: Data source is from the National Bureau of Statistics. This chart is the age-gender population pyramid from the seventh national census in 2020, showing the proportion distribution of men and women in different age groups. Symmetrical bar charts are used to visually reflect the population structure, such as the degree of aging and gender ratio differences. (A) X-axis: It represents the population ratio(%), with males on the left and females on the right, and the scale range is from (0.0% to 6.0%). (B) Y-axis: Age groups are divided by 10-year intervals (0-4 years old to 90-94 years old), arranged from the bottom (young population) to the top (elderly population).

the population structure in China. However, in some cases, extrinsic incentives may undermine intrinsic motivation, with opposing initial expected results [12]. Just as after the implementation of the two-child or even three-child fertility policy, the evolution of fertility intention shows a weakening trend [13]. This indicates that the marginal effect of relaxing the fertility quantity limit is decreasing [14], which should further analyze the reasons and mechanisms behind it, from macro to micro, and embed the individual decision-making process [15] in a broader social context.

From an economic point of view, the action mechanism of fertility willingness is consistent with the cost and utility theories [16]. On the one hand, parents raising children have two effects: one is to regard children as durable consumer goods, bringing happiness to their own, and other is) raise children as investment products,raising children to deal with the problem of old age care in the future [17]. However, having a child requires time and money from both parents, which is constrained by family income. Therefore, for a family, the decision to have several children can be analyzed from the perspective of parents' utility maximization. At the same time, giving birth to a child is in line with the law of diminishing marginal utility, that is, as the number of children increases, the utility that children can bring decreases. Referring to the study of Becker et al. [18], the following family fertility decision model was constructed:

$$Ut = U(c, m) + \rho V(u, w)$$

$U_t$ represents the total utility of the parent, $U(c,m)$ represents the current utility of the child as a durable consumer product, and $V(u,w)$ represents the expected utility of the child as an investment product. where C is the parents' own consumption, m is the number of

children, u is the cost that parents spend on one child, w is the level of social development, and is the discount factor, that is, the rate of return that parents expect their children to bring. To facilitate calculations, this study uses the Cobb-Douglas function and the natural log function to represent the current and expected utility of the parents and set the budget constraints that the parents face. The utility maximization problem of the parents is:

$$max: Ut = c^{\alpha}m^{\beta} + \rho ln(qu + w)$$

$$s.t. \quad \begin{cases} c + um = I \\ 0 < \alpha, \beta, \rho < 1 \end{cases} \quad (1)$$

where $\alpha, \beta$ is the preferences of parents, that is, the relative importance of parents to their own consumption and the number of children, q is the quality and potential of children, and I is parents' income. The first-order condition must be satisfied by using the Lagrange multiplier method.

$$\begin{cases} \frac{\partial U}{\partial c} = \alpha c^{\alpha-1}m^{\beta} = \lambda \\ \frac{\partial U}{\partial m} = \beta c^{\alpha}m^{\beta-1} = \lambda u \\ \frac{\partial U}{\partial u} = \frac{\rho q}{qu+w} = \lambda m \end{cases} \quad (2)$$

The optimal number of family decisions to have children:

$$m^* = \left( \frac{\rho q}{\alpha(qu + w)} \left( \frac{\alpha I}{\alpha + \beta} \right)^{1-\alpha} \right)^{\frac{1+\beta}{2}} \quad (3)$$

As can be seen from the above formula, the optimal number of parents is positively correlated with the product of parents' income I and their own consumption preference $\alpha$ and the quality of children q, and negatively correlated with the cost of parenting u and the degree of social development w. This indicates that parents are likely to choose to have fewer children to concentrate resources to produce higher-quality children [19], and with the continuous development of society, parents do not need to have more children. Although universal two-child and three-child policies are designed to encourage couples to have children actively, their fertility intentions and decisions depend more on the actual situation of each family. From a macro perspective, this study analyzes these factors affecting fertility decisions in a deeper way and considers the relationship between fertility policy and these factors.

## Study design

### Model construction

The comprehensive two-child or three-child policy is a nationwide universal policy, but the implementation of the birth policy and effect differences in different provinces and regions, in general, urbanization will affect the effect of birth policy adjustment and fertility level [20]; rural areas are sparsely combined with information spreads slowly, making birth policy on rural population constraints and stimulus relatively small [21]; in other words, from family planning to the comprehensive two-child and three-child policies, the urban population affected by the policy level is higher than the rural population. So this paper use the practice of Liang Zhihui [3], use of "urban population proportion" the continuous variables to distinguish the experimental group and control group, the continuous DID model is different from

the general setting virtual variable DID, but it will not change its basic properties, but can capture more data variability, and avoid the artificial set experimental group and control group may bring bias [22]. In view of this, this study selects panel data of 31 provinces, regions, and municipalities in China from 2005 to 2021 and constructs the following continuous DID model:

$$Y_{it} = \alpha_1 + \beta_1 Crate_{it} \times I_t^{post} + \phi X_{it} + \mu_i + \sigma_t + \varepsilon_{it} \qquad (4)$$

Among them, $Y_{it}$ represents the birth and fertility rates of i in year t, $Creat_{it}$ represents the proportion of urban population, $I_t^{post}$ represents the virtual variable representing the time point of 2016 as 1, otherwise 0; $X_{it}$ is a series of control variables affecting the birth rate and fertility; $\mu_i$ is a constant item, $\sigma_t$ representing the fixed effect of province, represents the fixed effect of year, and $\varepsilon_{it}$ represents the random disturbance term. The coefficient $\beta_1$ is the core explanatory variable of this study, which represents the net effect after the implementation of the universal two-child policy. If $\beta_1 > 0$ is significant, it indicates that the universal two-child policy promotes the birth and fertility rates; otherwise, it has an inhibitory effect; if $\beta_1$ not significant, it indicates that the implementation effect of the universal two-child policy is not obvious.

## Variable selection

The variables explained in this study are the birth and fertility rates of each province and city. Birth rate is the ratio of the number of births within the year to the total population at the end of the year, and fertility rate is the ratio of the number of births within the year to the average number of women of childbearing age. The two have the same numerators but different denominators. However, given that there are no statistical data on the number of women of childbearing age in China, this paper shows the fertility rate from the number of births in the year to the number of women aged 15 years and above. The core explanatory variable is the adjustment of the birth policy, representing the interaction term between the proportion of the urban population to the total population of the region and the virtual variable at the implementation time of the comprehensive two-child policy.

Considering the many factors affecting birth and fertility rates, a series of control variables was selected in this study. In addition, the adjustment of the birth policy is mainly affected by the population structure of China, so the proportion of children, proportion of the elderly, and sex ratio are mainly selected as the variables affecting the implementation of the universal two-child policyin the treatment effect model to test the robustness. Referring to the studies of Zhang Xiayu [23] and Su Liyun [24], the specific variables are listed in Table 1.

## Data description

The data used in this study are from the China Statistical Yearbook, China Population and Employment Statistical Yearbook, Statistical Yearbook of Provinces and Cities, and China Taian Database. The missing data in some provinces and cities were filled by finding the statistical bulletin and linear interpolation method of the province. Descriptive statistics for the main variables are presented in Table 2.

## Analysis of the empirical results

### Benchmark regression results

This study first examines the regression of dummy variables at the implementation point of the comprehensive two-child policy without adding interaction terms, and the results are

Table 1. **Main control variables and their definitions.**

| Bedding | Variable | Definition |
|---|---|---|
| Controlled variable | Economic development level | The actual GDP / region of each province / region total population (yuan / person), take logarithm |
| | Density of population | The average population density of the city (person / square kilometer), take the log |
| | Medical level | The number of health institutions owned by provinces and cities / the total population of the region (one / 10,000) |
| | Maternity insurance coverage rate | The number of people participating in maternity insurance in the provinces and cities / the total population of the region |
| | Education level | Average length of education = (population of college or above × 16 + high school × 12 + middle school × 9 + primary school × 6 + illiterate × 2) / number of population of 6 years or above (year / person), take logarithm |
| | Consumption level of residents | The total consumption of residents in each province and city / the total population of the region (yuan / person), take the log after CPI reduction |
| | Child dependency ratio | Population of children aged 0-14 / working age population (%) |
| | Senior dependency ratio | Population aged 65 years and above / working age population (%) |
| Population structure ($z_{it}$) | The proportion of children | Population of children aged 0 to 14 years (%) |
| | The proportion of old age | Population aged 65 years and older / total population (%) |
| | Sex ratio | The ratio of men and women in all provinces and cities (women = 100) |

Table 2. **Descricriptive statistics of the variables.**

| variable | observed value | mean | least value | crest value |
|---|---|---|---|---|
| Population birth rate | 527 | 11.009 | 3.59 | 17.94 |
| Fertility rate | 527 | 27.19 | 7.95 | 48.4 |
| The proportion of urban population | 527 | 0.5435 | 0.2071 | 0.896 |
| Economic development level | 527 | 9.682 | 8.528 | 10.872 |
| Density of population | 527 | 7.803 | 5.242 | 8.749 |
| Medical level | 527 | 1.689 | .214 | 3.08 |
| Maternity insurance coverage rate | 527 | 0.115 | 0.01 | 0.613 |
| Education level | 527 | 2.186 | 1.537 | 2.543 |
| Consumption level of residents | 527 | 9.4299 | 8.6876 | 10.4511 |
| Child support ratio | 527 | 23.879 | 9.64 | 44.65 |
| Senior support ratio | 527 | 13.969 | 6.71 | 26.7 |
| The proportion of children | 527 | 17.145 | 7.559 | 28.336 |
| The proportion of old age | 527 | 10.11 | 4.824 | 18.805 |
| sex ratio | 527 | 104.302 | 94.65 | 123.17 |

shown in Table 3 columns (1)–(2). It can be found that the estimated coefficient of the virtual variables is only significant on the birth rate at the level of 10%, and there is no control two-way fixed effect. Columns (3)–(6) report the urban population proportion and policy of the virtual variable interaction estimates for the birth rate and fertility rate. Even if the two-way fixed effect is not joined, the core estimation coefficient is only at the 10% level, and once the two-way fixed effect is not significant, it shows that a comprehensive two-child policy to improve the effect of the birth and fertility rates is generally minimal.

## Parallel trend test

The prerequisite for the use of the DID model is to meet the parallel trend hypothesis, which requires that the birth rate and fertility rate of the experimental and control groups change together before the implementation of the birth policy. Using the practices of Nathan [25],

**Table 3. Benchmark regression estimation results.**

| variable | (1) | (2) | (3) | (4) | (5) | (6) |
|---|---|---|---|---|---|---|
| $I^{post}$ | birth rate | fertility rate | birth rate | fertility rate | birth rate | fertility rate |
| | 0.4961* | 1.0030 | | | | |
| | (1.7373) | (1.3640) | | | | |
| Crate× $I^{post}$ | | | 0.8567* | 1.8678* | 1.9858 | 5.7728 |
| | | | (1.9159) | (1.6611) | (1.4121) | (1.6830) |
| Economic development level | −0.1231 | −1.1259 | −0.0937 | −1.0325 | 0.8745 | 0.7482 |
| | (−0.1834) | (−0.7267) | (−0.1394) | (−0.6674) | (0.9268) | (0.3128) |
| density of population | 0.0035 | 0.0780 | −0.0066 | 0.0585 | −0.0365 | 0.0995 |
| | (0.0124) | (0.1018) | (−0.0231) | (0.0759) | (−0.1061) | (0.1053) |
| medical level | 0.5634*** | 1.3824*** | 0.5766*** | 1.4217*** | 0.5659 | 1.7779 |
| | (2.6757) | (2.6405) | (2.7515) | (2.7167) | (1.3230) | (1.6445) |
| Maternity insurance coverage rate | 1.6885 | 7.0096 | 0.9927 | 5.4643 | 4.4624** | 12.7089*** |
| | (0.7718) | (1.5059) | (0.4079) | (1.0443) | (2.5825) | (3.1326) |
| Education level | −7.4530*** | −17.5741*** | −7.4024*** | −17.6279*** | 4.7400 | 11.0383 |
| | (−5.0339) | (−4.2671) | (−5.0862) | (−4.3304) | (1.2911) | (1.1403) |
| Household consumption level | 0.0836 | 0.7540 | 0.1321 | 0.7885 | −1.1370 | −1.6383 |
| | (0.1228) | (0.4230) | (0.1861) | (0.4240) | (−0.8404) | (−0.4239) |
| Child dependency ratio | 0.0859** | 0.4091*** | 0.0839** | 0.4008*** | 0.0773* | 0.3592*** |
| | (2.3240) | (4.7744) | (2.2802) | (4.6806) | (1.8579) | (3.2933) |
| Senior dependency ratio | −0.2935*** | −0.7991*** | −0.2980*** | −0.8117*** | 0.0449 | 0.0394 |
| | (−7.1571) | (−8.0207) | (−7.1927) | (−8.0074) | (0.5417) | (0.1772) |
| constant term | 28.4054*** | 66.6854*** | 27.7891*** | 66.1655*** | 0.3581 | 0.3461 |
| | (3.7119) | (3.5568) | (3.5231) | (3.4129) | (0.0181) | (0.0063) |
| Province fixed effect | deny | deny | deny | deny | yes | yes |
| Year fixed effect | deny | deny | deny | deny | yes | yes |
| N | 527 | 527 | 527 | 527 | 527 | 527 |
| R² | 0.6631 | 0.7836 | 0.6587 | 0.7792 | 0.6057 | 0.5824 |

Note: Steady t-values are clustered at the province level in brackets, and *, * *, and * * * indicate significance at the 10%, 5%, and 1% levels, respectively.

Beck [26], etc., the following model is constructed to test the parallel trend hypothesis:

$$Y_{it} = \alpha_1 + \sum_{t=-2005}^{2021} \beta_t Crate_{it} \times D_t + \varphi X_{it} + \mu_i + \sigma_t + \varepsilon_{it}$$

Among them, $D_t$ represents the year dummy variable, and the other variables and coefficients are consistent with the benchmark regression. The test results are shown in Fig 4. Five years before the policy was launched, the trends of the experimental group and the control group were consistent. It seems that the universal two-child policy has no significant impact on the birth rate and fertility rate of the urban population, which is in line with the parallel trend hypothesis. Further analysis of its dynamic effect shows that after the policy was implemented, the estimated value of the coefficient points gradually increased, but the confidence interval still contained zero values. The statistical insignificance indicates that this difference might be a random fluctuation rather than the result of policy intervention.

## Placebo test

**Time placebo test.** This study employs a placebo test design along the time dimension and conducts a robustness test on the implementation effect of the universal two-child policy based on the counterfactual framework. Specifically, by systematically advancing the policy intervention time point in the pilot areas by 1 to 10 years, virtual treatment groups

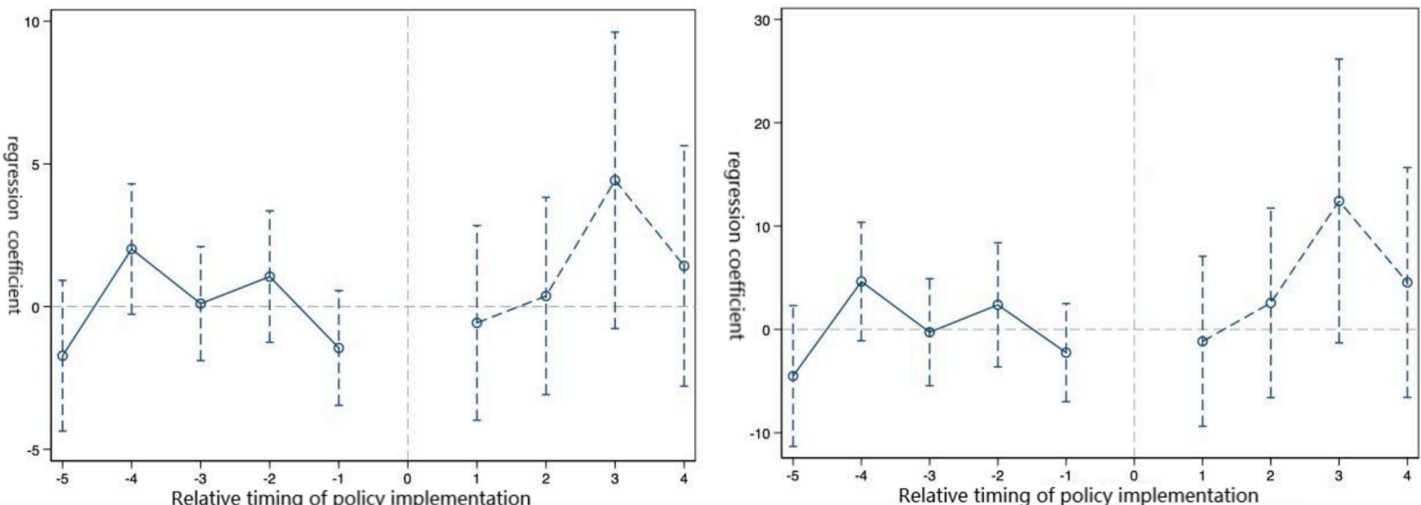

A) The outcome variable is the population birth rate (b) The outcome variable is the fertility rate.

**Fig 4. Parallel trend test.** Note: (A) The outcome variable is the population birth rate (B) The outcome variable is the fertility rate.The parallel trend test results in Fig 4 show that before the policy implementation, the regression coefficients of the birth rate (Fig A) and the fertility rate (Fig B) fluctuated around zero, indicating that the trends were basically parallel before the policy implementation. After the policy was implemented (with a positive relative time), the regression coefficient increased significantly and reached a peak at a relative time of 3, and then decreased slightly. This indicates that the policy has had a significant positive impact on the birth rate and fertility rate of the population. Therefore, it can be concluded that the policy has effectively increased the birth rate and fertility rate of the population.

are constructed to generate placebo policy variables. Under the setting of the difference-in-differences model (DID), if the estimated coefficients of the advanced policy time points are statistically significant ($p < 0.1$), it indicates that the observed changes in the birth rate and fertility rate may be due to potential confounding factors rather than the effect of the target policy; conversely, if the estimated coefficients cannot reject the null hypothesis at the traditional significance level, it effectively rules out the interference effect of other exogenous policy shocks on fertility behavior. The results of the placebo test shown in Fig 5. indicate that when the time point of the policy is advanced to three years before the actual implementation year, the estimated coefficients of the main explanatory variables do not pass the 10% significance level test (the 95% confidence interval includes zero). The empirical results meet the conditions for conducting a placebo test on the implementation time of the universal two-child policy, further confirming the robustness of the benchmark regression results.

**Random treatment group placebo test.** To avoid the influence of potential omitted variable bias on causal inference, a non-parametric permutation test method is used to construct a counterfactual analysis framework. The randomized interaction term is included in the regression equation for regression. By randomly permuting the interaction term of the treatment group dummy variable and the time dummy variable 500 times, a placebo effect reference distribution is generated. In each simulation, the randomized treatment effect term is substituted into the benchmark regression equation to obtain the virtual policy effect coefficient and its significance level. Based on the simulation results, a kernel density plot of the coefficient estimates and a cumulative distribution function (CDF) plot of the p-values are drawn. The empirical results show (Fig 6): First, the estimated coefficients of the false regression are far from the estimated values of the benchmark regression. Second, the regression coefficients are close to zero and approximately follow a normal distribution, and most of the p-values of the estimated coefficients are above 0.1, indicating that the fertility policy has no

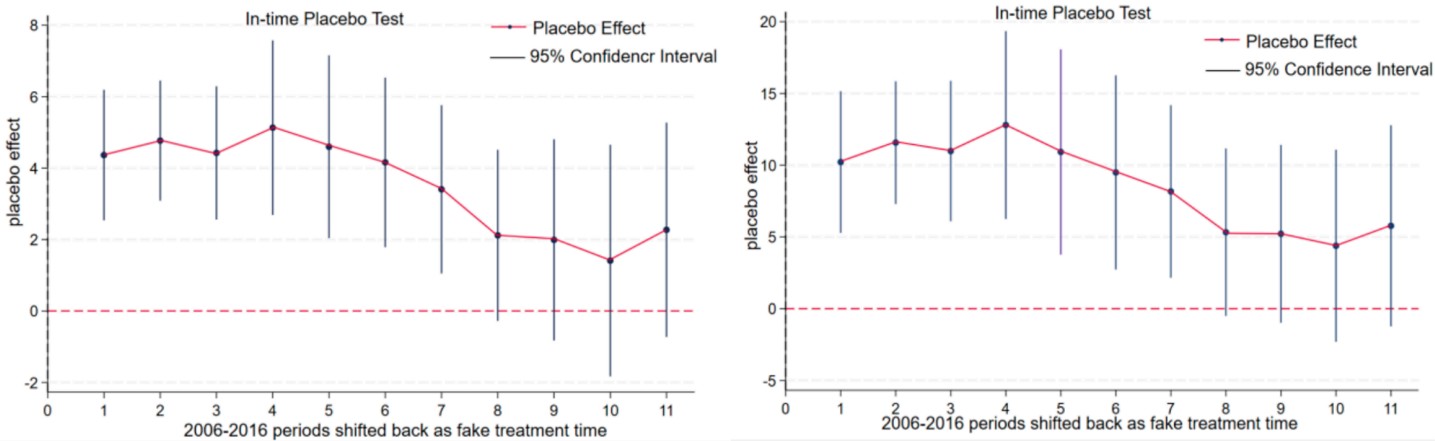

A) The outcome variable is the population birth rate (b) The outcome variable is the fertility rate.

**Fig 5. 2006-2016 periods shifted back as fake treatment time.** Note: (A) The outcome variable is the population birth rate (B) The outcome variable is the fertility rate. This graph verifies the parallel trend hypothesis of the birth rate and fertility rate before the intervention by moving back the time axis of the data from 2006 to 2016 to simulate the fictional processing time.

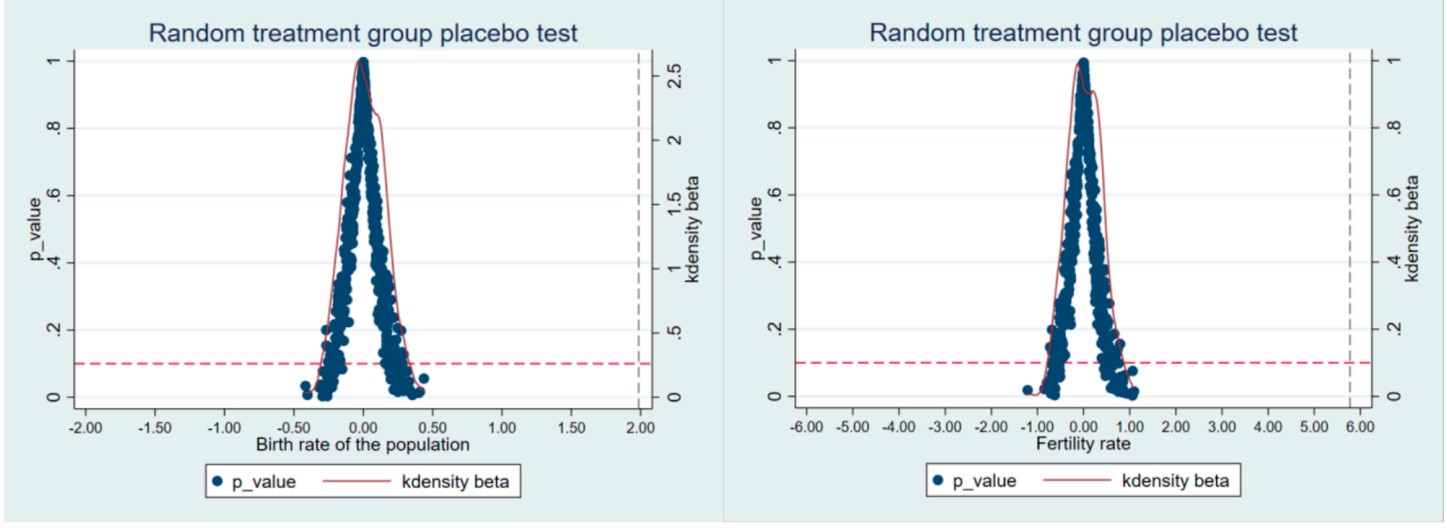

A) The outcome variable is the population birth rate (b) The outcome variable is the fertility rate.

**Fig 6. Random treatment group placebo test.** Note: (A) The outcome variable is the population birth rate (B) The outcome variable is the fertility rate. The placebo test confirmed that the policy effects observed in the benchmark regression were not caused by omitted variables or random fluctuations, and the conclusion was reliable.

significant effect in 500 random samplings, proving that the benchmark regression results of this paper are relatively robust.

## Robustness test

**Treatment effect model.** In view of the effect evaluation of fertility policy, there are endogenous problems such as reverse causality and selection bias; that is, the birth rate will,

in turn, affect the choice of the implementation time of fertility policy, so the treatment effect model (TEM) in this study can alleviate its endogeneity to some extent. The treatment effect model (TEM) was proposed by Maddala in 1983 to follow the Heckman sample selection model and directly perform structural modelling of the processing variable $D_{it}$. details are as follows:

$$y_{it} = \varphi x_{it} + \gamma D_{it} + \varepsilon_{it}$$

In the above equation, $x_{it}$ is a series of directly affected control variables that directly affect $y_{it}$, $\varepsilon_{it}$ represents the random disturbance terms, and the treatment variables $D_{it}$ are determined by the following "treatment equation":

$$D_{it} = \tau(z_{it}\delta + u_{it})$$

Among them, $\tau(\cdot)$ is a schematic function, $z_{it}$ can be regarded as a tool variable of $D_{it}$, which indirectly influence $y_{it}$ by influencing $D_{it}$. The treatment effect model is similar to the two-step estimation of Heckit, estimated $\widehat{\delta}$ with Probit, and calculated $\widehat{\lambda}_i$ with inverse Mills ratio, then substituting in toto obtain the estimate using OLS regression. However, this method introduces the estimation error of the first step into the second step, and the two-step method cannot test whether the model has an endogenous problem [27]. Therefore, a more efficient approach is to use the maximum likelihood estimation method (MLE) to estimate all model parameters simultaneously. Table 4 reports the estimated results of the treatment effect model. Since child dependency ratio and old-age dependency ratio have a serious linear correlation with demographic structure $z_{it}$, they were removed from the control variables. Column (1) and (3) were estimated by the two-step method, and column (2) and (4) were estimated by the maximum likelihood estimation method. Both hazard and Wald endogeneity tests rejected the original hypothesis at the 1% significance level, indicating that the treatment effect model can obtain unbiased effective estimation [28]. But no matter what kind of estimation method, comprehensive two child policy implementation of the birth rate and fertility rate have negative significant effect, this is a bit unexpected, the reason is that after the comprehensive two child implementation, the birth rate has been falling, and comprehensive two children is "one size fits all" policy, processing effect model here cannot control year fixed effect, so the estimated coefficient is significantly negative, but this confirmed from the side the rationality of the continuous DID model, also verified the comprehensive two child policy little effect. In addition, the proportion of middle-aged and elderly in the population structurehas the greatest impact on the virtual variables of the universal two-child policy [29], indicating that the aging of the population is one of the major bases for the adjustment of China's birth policy, which is in line with the actual national conditions.

**Change of the explained variable.** To study the expected effect of the birth policy adjustment more comprehensively, this study replaces the explanatory variables with the number of births (absolute number) and the natural population growth rate, that is, the difference between the birth rate and the mortality rate. As shown in Table 5 columns (1) and (2), the comprehensive two-child policy to improve the birth number base has no significant effect, but the natural population growth rate of 10% is significant; this may be because with the development of society, peoples' life expectancy generally increases, mortality continues to decrease, and makes the natural population growth rate relative rise. However, in general, the implementation of the universal two-child policy has not reached expectations.

**Lag regression.** As the saying goes, "October is pregnant," which takes nearly a year from conception to the birth of the baby. Therefore, there will be a lag effect in the implementation of the birth policy, and it will take time to take effect. To this end, all explanatory variables

**Table 4. Presents the estimation results of the effect model.**

| variable | (1) | (2) | (3) | (4) |
|---|---|---|---|---|
| | birth rate | birth rate | fertility rate | fertility rate |
| $I^{post}$ | −2.1893*** | −1.2137*** | −4.6889*** | −2.7485*** |
| | (−6.0878) | (−5.0469) | (−5.2973) | (−4.3920) |
| Economic development level | 0.1947 | 0.4909 | −1.5238 | −0.9058 |
| | (0.2963) | (0.7466) | (−0.9230) | (−0.5494) |
| density of population | −0.1886 | −0.2030 | −0.6170 | −0.6449 |
| | (−0.8837) | (−0.9615) | (−1.1568) | (−1.2187) |
| medical level | 0.6116*** | 0.6484*** | 1.5208*** | 1.5892*** |
| | (3.1742) | (3.3527) | (3.1444) | (3.2733) |
| Maternity insurance coverage rate | 3.8946** | 3.1109* | 11.4534** | 9.8520** |
| | (2.1591) | (1.7115) | (2.5243) | (2.1635) |
| Education level | −10.5548*** | −10.3693*** | −29.7210*** | −29.2641*** |
| | (−4.6022) | (−4.4084) | (−5.1145) | (−4.9495) |
| Household consumption level | −0.1504 | −0.3958 | 0.1885 | −0.3333 |
| | (−0.2923) | (−0.7436) | (0.1434) | (−0.2486) |
| constant term | 33.0270*** | 31.9809*** | 105.0737*** | 102.8340*** |
| | (4.6463) | (4.4723) | (5.8740) | (5.7232) |
| The proportion of children | 0.1002*** | 0.1219*** | 0.1002*** | 0.1143*** |
| | (5.3959) | (6.6320) | (5.3959) | (6.2534) |
| The proportion of old age | 0.4214*** | 0.4304*** | 0.4214*** | 0.4274*** |
| | (11.6473) | (13.0075) | (11.6473) | (12.6316) |
| sex ratio | 0.0738*** | 0.0651*** | 0.0738*** | 0.0572*** |
| | (4.2987) | (3.9309) | (4.2987) | (3.3509) |
| constant term | −14.1422*** | −13.6685*** | −14.1422*** | −12.6876*** |
| | (−7.2266) | (−7.3425) | (−7.2266) | (−6.6239) |
| $\lambda/chi^2(p)$ | 1.4971*** | 34.32*** | 3.1086*** | 22.68*** |
| | (6.7594) | (0.0000) | (5.6470) | (0.0000) |
| Province fixed effect | yes | yes | yes | yes |
| N | 527 | 527 | 527 | 527 |

Note: Steady t-values are clustered at the province level in brackets, and *, * *, and * * * indicate significance at the 10%, 5%, and 1% levels, respectively.

**Table 5. The estimation results from the robust test.**

| variable | (1) | (2) | (3) | (4) |
|---|---|---|---|---|
| | Birth number | rate of natural increase | birth rate | fertility rate |
| $Creat*I^{post}$ | 0.1686 | 2.2814* | 0.9780 | 3.3736 |
| | (1.3221) | (1.8729) | (0.6695) | (0.8767) |
| controlled variable | yes | yes | yes | yes |
| Province fixed effect | yes | yes | yes | yes |
| Year fixed effect | yes | yes | yes | yes |
| N | 527 | 527 | 496 | 496 |
| R² | 0.6007 | 0.7238 | 0.6169 | 0.5802 |

were delayed by one year before the regression. The results are shown in Table 5 (3) and (4). The estimated coefficient is still positive and insignificant, which shows that with the passage of time, the universal two-child policy has not significantly improved the fertility level of the provinces, which again proves that the universal two-child policy has not significantly affected the birth and fertility rates in China.

**Eliminate interference factors.** The possible decline in fertility willingness due to the impact of COVID-19 in 2020 and 2021 should be considered. To avoid the interference

caused by the epidemic factors, this study eliminated the data of the affected two years and reduced the time window to 2005-2019. As shown in columns (1) and (2) in Table 6, the estimated coefficient is still not significant, which is consistent with the benchmark regression results. This further confirms the robustness of the core conclusion of this study, that is, the universal two-child policy has little effect.

**The Comprehensive three-child policy.** In 2021, to further optimize the birth policy, China has relaxed that a couple can have two to three children in order to maintain the country's advantage in human resource endowment. Therefore, this study preliminarily studies the implementation effect of the universal two-child to three-child policy, uses the data from 2016 to 2021, assigns the value of 2021 to 1 and 0 in the other years, and multiplies it by the proportion of urban population. Table 6 columns (3) and (4) show the impact of the implementation of the comprehensive three-child policy on the birth and fertility rates in China. The results are still positive and insignificant, but part of the reason is the lack of data after the comprehensive three-child policy in 2021, which may cause a certain deviation in the estimated results.

## Heterogeneity analysis

In view of the differences in location advantages, economic level, population quality, and ideology among various regions in China, and the corresponding heterogeneity in the implementation effect of the universal two-child policy in different regions, the provinces and municipalities were divided into 11 provinces in the eastern region, 10 provinces in the central region, and 10 provinces in the western region [30]. Table 7 shows the estimated results of the birth rate as the explained variable, and columns (1) to (3) show that the implementation effect of the universal two-child policy is largest in the central region and significant at the 10% level. The reason may be that the previous family planning to the east and central economy developed relatively more strictly, so comprehensive two-child policy once, these areas fertility enthusiasm, but also due to the eastern coastal provinces of urbanization and life pressure, so the policy effect in the east is not obvious, but played a role in the central region. However, the implementation of family planning for ethnic minorities and rural areas is relatively loose, and the geographical and population conditions in the western region may make no obvious difference between the implementation of the universal two-child policy before and after implementation. In other words, the fertility level in the western region did not fluctuate significantly with the implementation of the universal two-child policy [31]. To further confirm the above reasons, this study grouped the five ethnic minority autonomous regions into Yunnan and Guizhou provinces, which have a relatively large number of ethnic minorities, and the other 24 provinces and cities. The results are shown in columns (4) and (5). Although neither is significant, the estimated coefficient for the remaining 24 provinces

**Table 6. Other factors estimate the results.**

| variable | (1) | (2) | (3) | (4) |
|---|---|---|---|---|
| | birth rate | fertility rate | birth rate | fertility rate |
| $Creat*I^{post}$ | 1.7227 | 5.2088 | 1.5244 | 3.1716 |
| | (1.1135) | (1.3592) | (0.9193) | (0.6586) |
| controlled variable | yes | yes | yes | yes |
| Province fixed effect | yes | yes | yes | yes |
| Year fixed effect | yes | yes | yes | yes |
| N | 465 | 465 | 186 | 186 |
| R² | 0.1912 | 0.2422 | 0.7478 | 0.7143 |

**Table 7. Analysis of the heterogeneity.**

| variable | (1) | (2) | (3) | (4) | (5) |
|---|---|---|---|---|---|
| | east | central section | west | municipality | Other provinces and cities |
| $Creat*I^{post}$ | 0.6622 | 6.8319* | 2.3040 | -1.9038 | 1.7010 |
| | (0.2410) | (2.0603) | (1.1840) | (-0.2975) | (0.9871) |
| controlled variable | yes | yes | yes | yes | yes |
| Province fixed effect | yes | yes | yes | yes | yes |
| Year fixed effect | yes | yes | yes | yes | yes |
| N | 187 | 170 | 170 | 119 | 408 |
| R² | 0.6319 | 0.7569 | 0.5527 | 0.5316 | 0.6769 |

and cities is positive. In summary, there is some heterogeneity in the implementation of the universal two-child policy.

## Identification test and cause analysis

The above analysis shows that the comprehensive two-child policy and even the comprehensive three-child policy have not significantly improved birth and fertility rates in China. In other words, the effect of adjusting the birth policy was a drop in the ocean. Why do the universal two-child and three-child policies encourage people to have children, but they fail to play their due role? To explore the reasons behind this, this paper further examines the action mechanism of fertility policy and draws the research of Feng Feng [32], Jiangboat [33], etc. The specific model was set as follows:

$$M_{it} = \alpha_2 + \beta_2 policy_t + X_{it}'\xi + \mu_i + \sigma_t + \varepsilon_{it}$$

Among them, $X_{it}'$ represents a set of control variables, including economic development level, population density, medical care level, education level, and household consumption level, and the meaning of the other symbols is consistent with the basic model. The relaxation of the birth policy is mainly based on peoples' willingness to have children, improving the fertility level in China by improving fertility intention; however, fertility intention is a subjective consciousness that is difficult to measure directly. Therefore, this study takes the Baidu index as a breakthrough point. [34], As shown in Fig 5, Select the key words "pregnancy preparation," "pregnancy preparation precautions," Manual organizing the overall daily average of the two keywords in the provinces and cities in the past decade, Divide it by the total population at the end of the year, It can reflect the fertility will of the provinces to some extent, It can be found that the changing trends of these two keywords are generally consistent with the population birth rate in Fig 2, Show that the selected keywords are reasonable and effective, Can reflect changes in fertility intentions, Control variables from Internet penetration rate and benchmark regression are also added here, So as to better judge whether the universal two-child policy can improve the fertility intention. The results are shown in Table 8 (1) and (2), where the universal two-child policy has not significantly improved people's willingness to have children, and it is difficult to substantially improve the fertility level in China.

As can be seen from Fig 7, from 2011 to 2021, the search indices for "preparing for pregnancy" and "pregnancy precautions" decreased by 67% and 60% respectively, and the long-term trend has continued to decline. From the perspective of key nodes, after the implementation of the universal two-child policy in 2016, the search index did not show the expected

**Table 8**. Identification test and cause analysis.

| variable | (1) get ready for pregnancy | (2) Precautions for pregnancy preparation | (3) Engel coefficient | (4) CPI | (5) employment rate | (6) Birth participation rate |
|---|---|---|---|---|---|---|
| *Creat*I^{post}* | −0.0396 | 0.0019 | −0.0043 | −5.4127 | 0.0894* | 0.2402*** |
| | (−1.5895) | (0.2843) | (−0.1965) | (−1.6723) | (1.9049) | (3.2222) |
| controlled variable | yes | yes | yes | yes | yes | yes |
| Province effect | yes | yes | yes | yes | yes | yes |
| Year effect | yes | yes | yes | yes | yes | yes |
| N | 341 | 310 | 527 | 527 | 475 | 527 |
| R² | 0.6836 | 0.7720 | 0.9003 | 0.9828 | 0.2428 | 0.7005 |

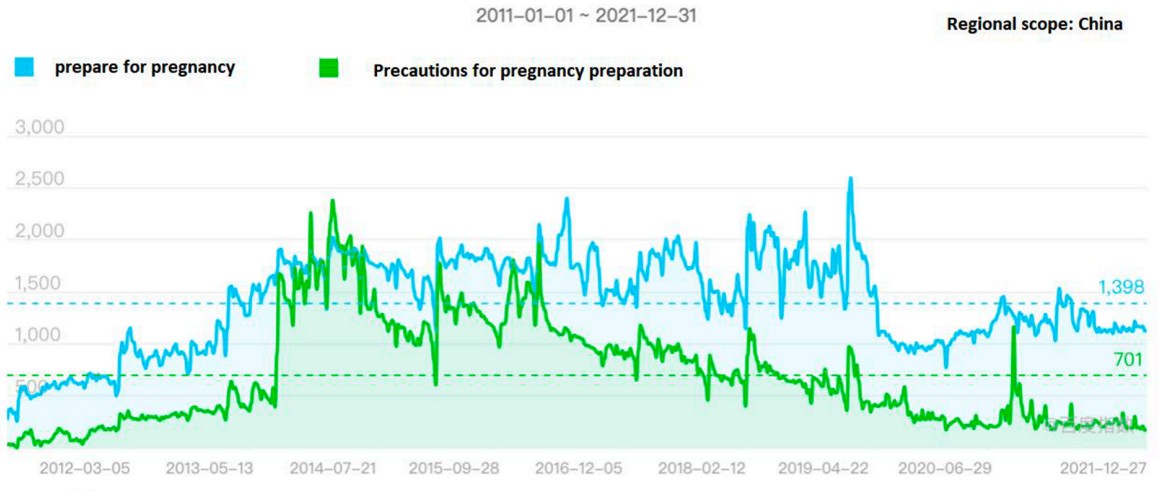

**Fig 7. Search index of "Preparation for pregnancy" and "precautions for pregnancy preparation" in 2011-2021.** Note: Data source baidu index official website.This chart shows the trend of Baidu search index for "prepare for pregnancy" and related precautions in China from 2011 to 2021, reflecting the changes in public attention to the topic of preparing for pregnancy. This chart is based on Baidu Index and presents the changes in search popularity among Chinese users for "preparing for pregnancy" and its precautions from 2011 to 2021. Key dates may correspond to policy releases or social phenomena.

rebound but continued to decline, verifying the low correlation between the policy effect and the willingness to have children. During the COVID-19 pandemic from 2020 to 2021, the search index further dropped to its lowest point, reflecting the inhibitory effect of intensified economic uncertainty on fertility decisions. In the long term, the continuous decline in the search index indicates that policy relaxation is difficult to counteract the suppression of fertility intentions by structural factors such as the transformation of marriage and childbearing concepts and the increase in opportunity costs. In addition, the Baidu index only reflects a part of fertility intention, and there are many reasons why fertility policies are not effective. The fertility level in China is the result of a combination of multiple factors, so this study combines the fertility decision-making model of theoretical analysis to explore the relationship between fertility policy and these factors to better explore the effective internal mechanism of unsuccessful fertility policy. Specifically, these factors that affect fertility decisions can be summarized into two parts: unwillingness to have children (willingness) and reluctance to have children (ability): the improvement of material life and education level leads to a change

in marriage and childbearing concepts, and at the same time, couples' preference for social pension security will have a crowding out effect on their willingness to have multiple children, that is, "unwillingness to have children"; However, due to the influence of previous fertility policies, as well as the high opportunity cost of raising children and the employment pressure on women of childbearing age, the marriage and childbearing age group is unable to bear the pressure of having more children, that is, "dare not have children." (1) Change in the fertility concept. The economic foundation determines the superstructure, with the improvement of peoples' material living standards, and the concept of fertility may change. In the past, due to the backward economy, people hoped to ensure the labor force of the family population. However, with the rapid development of society, the social wealth creation of the unit working population has been far beyond the past, and people no longer urgently need to solve their old-age problems through fertility and ensure the operation of the family economy. That is, they experienced a change from the previous raising of children for old age to the later social pension, self-reliance. Therefore, this study measures the living standards of families using the Engels' coefficient. As shown in Table 8 (3), the implementation of the universal two-child policy does not affect the Engels coefficient, which means that the birth policy is not closely related to people's living standards; therefore, it is difficult to prevent changes in the concept of childbearing. (2) The economy faces downward pressure. In recent years, the international and domestic situation is changing rapidly, China's economic structure is in a state of adjustment, the material level of the cost of living is also increasing, people's expectations for the future are low, coupled with the current children's education, medical care, and other parenting costs are high, and the problem of "can't afford to be raised" is increasingly obvious. The rising price level year by year and grim economic situation seriously affect couples' children. Many families choose not to have children if they cannot clearly give their children a better future. Here, the CPI index is used to measure the price level (taking 2005 as the base period), and the results are shown in Table 8, column (4). The universal two-child policy cannot significantly reduce the CPI, which indicates that people face high price levels and economic pressure, and their fertility intention is naturally significantly reduced. (3) The support measures of birth policy are not sufficiently perfect. Compared with the universal three-child policy, the support measures for the comprehensive two-child policy are not perfect, and the implementation is not sufficient. For a family,childbirth means a reduction in income sources and job opportunities, especially for women of childbearing age, not only facing discrimination in job hunting but also possible unemployment during pregnancy. However, adjustment of birth policy does not pay full attention to the employment and unemployment of women of childbearing age. On the contrary, the relaxation of the policy intensifies gender discrimination [35] in the labor market, and the risk of having children to bring to the family is greater, thus reducing fertility intention. Since there are no data on the employment of the female population in provinces and cities, the ratio of the employed population to the total population at the end of the year is used to represent the employment rate, and it is taken as an interpreted variable. The results are shown in (5) in Table 8, which shows that the implementation of the universal two-child policy has a limited effect on improving the employment rate. However, the universal two-child policy has significantly increased maternity insurance coverage. As shown in column (6), this is also one of the important reasons why maternity policy can play a role in some regions, but it is relatively insignificant. Overall, the universal two-child policy is ineffective.

(4) The existing dependency burden is high. The so-called "the sandwich generation", with the deepening of Chinas aging degree, provide for the aged has become a big problem, the pressure of the family is increasing [36], not to mention some families have already had a child, to have a second or even a third child, will undoubtedly further increase the pressure

of both husband and wife. Thus, with the old dependency ratio and total dependency ratio regression, the results are shown in Table 9 column (1) and (2), respectively. After the implementation of the comprehensive two-child policy, the burden of raising was not reduced, but the elderly dependency ratio at the level of 10% positive, it has a squeezing effect on fertility, indirectly transferring the family's energy on fertility, which leads to a decline in fertility.

(5) Dilution effect of the previous decision policy. The implementation of the universal two-child policy has not been achieved overnight, and relevant policies have been implemented. In the 21st century, Chinese control over family planning has weakened, and many rural families have already given birth to two or even three children. Before the introduction of the universal two-child policy, the country gradually opened the "double only two child policy" and "restricted two-child policy" birth policies. By 2016, with the implementation of the universal two-child system, their enthusiasm for childbearing decreased to a certain extent. Therefore, this study advances the policy time point to the implementation of the restricted two-child policy in 2013. The results are presented in (3) and (4) in Table 9. The significance of the estimated coefficient is improved compared to the benchmark regression, indicating that the restricted two-child policy has a certain dilution effect on the implementation effect of the universal two-child policy.

(6) Lower marriage and better divorce rates. In addition to social and economic factors, low marriage rate is an important reason of family planning in the 1980s, because each family can only have a child and the influence of "son preference" thought, cause serious imbalance, this on the one hand reduces the comprehensive two child period of marriageable base, on the other hand the general improvement of education level makes late marriage late childbearing become the norm. While the marriage rate is gradually declining, the divorce rate is rising — people's ideas, the acceleration of the pace of life, and the complexity of family contradictions–the turbulence of marriage causes the fertility rate to decline further. As shown in the estimated results of (5) and (6) listed in Table 9, the implementation of the universal two-child policy has not significantly affected marriage and divorce rates in China, and it is naturally difficult to affect fertility levels.

## Birth number forecast

Before this chapter, we analyzed the reasons why it is difficult to effectively adjust China's fertility policy. Further, this study starts with the adjustment of the universal two-child policy to the comprehensive three-child policy, the births from 2015 to 2021 are selected as the original data, and the GM (1,1) grey prediction model is applied to predict the births in the next five

**Table 9. Identification test and cause analysis.**

| variable | (1) Senior support ratio | (2) Total dependency ratio | (3) birth rate | (4) fertility rate | (5) Marriage rate | (6) divorce rate |
|---|---|---|---|---|---|---|
| $Creat*I^{post}$ | 4.8128* | 4.4061 | 1.9424* | 5.3278* | −4.2804 | 0.4024 |
| | (1.8403) | (1.2720) | (1.8352) | (1.8669) | (−1.4075) | (0.4434) |
| controlled variable | yes | yes | yes | yes | yes | yes |
| Province effect | yes | yes | yes | yes | yes | yes |
| Year effect | yes | yes | yes | yes | yes | yes |
| N | 527 | 527 | 527 | 527 | 527 | 527 |
| $R^2$ | 0.7939 | 0.7228 | 0.6062 | 0.5823 | 0.5893 | 0.7733 |

years of 31 provinces and cities. The GM (1,1) model is a univariate first-order gray prediction, which predicts future unknown data from small samples with partially known information and is suitable for short-term population prediction [37]. The specific establishment process is as follows. First, the raw data sequence $X^{(0)} = \{X^{(0)}(1), X^{(0)}(2), \cdots, X^{(0)}(n)\}$, the new sequence $X^{(1)} = \{X^{(1)}(1), X^{(1)}(2), \cdots, X^{(1)}(n)\}$ is generated by accumulation, and the newly generated sequence $X^{(1)}$ is used to fit the function curve. Then, the predicted value sequence of the newly generated sequence $X^{(1)}$ is obtained using the fitted function $X^{(1)}$; Finally, the reduction $X_{(0)}(k) = X_{(1)}(k) - X_{(1)}(k-1)$ reduction is used to obtain the gray predicted value sequence: $X_{(0)} = \{X_{(0)}(1), X_{(0)}(2), \cdots, X_{(0)}(n+m)\}$ [38]. The GM (1,1) grey forecast results are shown in Table 10, where the provinces and cities after five years of birth number are declining. The uncertainty due to the influence of the outbreak makes the forecast of 2020-2022 than the actual value, but also even if our country quickly gets rid of the negative impact from the outbreak, does not mean that the birth number will exhibit retaliatory growth. The last row of Table 10 is based directly on the prediction of the national birth number from 2015

**Table 10. Grey forecast results of birth number (ten thousand people).**

| predicted value | 2015 | 2016 | 2017 | 2018 | 2019 | 2020 | 2021 | 2022 | 2023 | 2024 | 2025 | 2026 |
|---|---|---|---|---|---|---|---|---|---|---|---|---|
| Beijing Municipality | 17.21 | 20.98 | 19.81 | 18.71 | 17.67 | 16.69 | 15.76 | 14.88 | 14.06 | 13.27 | 12.54 | 11.84 |
| Tianjin Municipality | 8.95 | 12.33 | 11.19 | 10.16 | 9.22 | 8.37 | 7.60 | 6.90 | 6.27 | 5.69 | 5.17 | 4.69 |
| Hebei Province | | 100.56 | 89.91 | 80.39 | 71.88 | 64.27 | 57.46 | 51.38 | 45.94 | 41.08 | 36.73 | 32.84 |
| Shanxi International | 36.49 | 41.22 | 38.00 | 35.03 | 32.30 | 29.78 | 27.45 | 25.31 | 23.34 | 21.51 | 19.84 | 18.29 |
| Gong and Drum Festival Nei Monggol | | 24.44 | 22.41 | 20.54 | 18.83 | 17.26 | 15.82 | 14.50 | 13.29 | 12.18 | 11.17 | 10.24 |
| Liaoning Province | 27.10 | 30.45 | 28.42 | 26.51 | 24.74 | 23.08 | 21.54 | 20.10 | 18.75 | 17.49 | 16.32 | 15.23 |
| Jilin Province | | 20.34 | 18.08 | 16.08 | 14.30 | 12.71 | 11.31 | 10.05 | 8.94 | 7.95 | 7.07 | 6.28 |
| Heilongjiang Province | 22.87 | 25.51 | 21.84 | 18.70 | 16.00 | 13.70 | 11.73 | 10.04 | 8.60 | 7.36 | 6.30 | 5.39 |
| Shanghai Municipality | 18.52 | 22.13 | 19.70 | 17.53 | 15.61 | 13.89 | 12.37 | 11.01 | 9.80 | 8.73 | 7.77 | 6.91 |
| Jiangsu Province | 72.11 | 83.33 | 75.81 | 68.96 | 62.73 | 57.06 | 51.91 | 47.22 | 42.95 | 39.07 | 35.54 | 32.33 |
| Zhejiang Province | 58.10 | 67.93 | 63.27 | 58.92 | 54.87 | 51.11 | 47.60 | 44.33 | 41.28 | 38.45 | 35.81 | 33.35 |
| Anhui Province | 79.38 | 88.84 | 79.31 | 70.79 | 63.19 | 56.41 | 50.35 | 44.95 | 40.12 | 35.82 | 31.97 | 28.54 |
| Fujian Province | 53.13 | 60.32 | 54.53 | 49.29 | 44.56 | 40.28 | 36.42 | 32.92 | 29.76 | 26.90 | 24.32 | 21.99 |
| Jiangxi Province | 60.10 | 67.00 | 60.28 | 54.23 | 48.79 | 43.89 | 39.49 | 35.52 | 31.96 | 28.75 | 25.87 | 23.27 |
| a folk art form popular | 143.00 | 173.46 | 149.25 | 128.42 | 110.50 | 95.08 | 81.81 | 70.39 | 60.56 | 52.11 | 44.84 | 38.58 |
| in Shandong Henan Province | 136.00 | 150.52 | 135.45 | 121.89 | 109.68 | 98.70 | 88.82 | 79.92 | 71.92 | 64.72 | 58.24 | 52.41 |
| Hubei province | 62.65 | 77.41 | 68.63 | 60.85 | 53.95 | 47.83 | 42.40 | 37.59 | 33.33 | 29.55 | 26.20 | 23.23 |
| Hunan Province | 91.80 | 98.69 | 85.82 | 74.63 | 64.89 | 56.43 | 49.07 | 42.67 | 37.10 | 32.26 | 28.06 | 24.40 |
| Guangdong Province | 119.95 | 146.01 | 144.40 | 142.81 | 141.24 | 139.68 | 138.15 | 136.63 | 135.12 | 133.64 | 132.17 | 130.71 |
| Guangxi | | 82.42 | 75.27 | 68.74 | 62.77 | 57.32 | 52.35 | 47.80 | 43.65 | 39.87 | 36.41 | 33.25 |
| Hainan Province | 13.27 | 14.15 | 13.32 | 12.54 | 11.81 | 11.12 | 10.47 | 9.86 | 9.28 | 8.74 | 8.23 | 7.75 |
| Chongqing City | 37.34 | 39.12 | 34.86 | 31.07 | 27.69 | 24.68 | 21.99 | 19.60 | 17.47 | 15.57 | 13.88 | 12.37 |
| Sichuan Province | 84.00 | 95.66 | 88.11 | 81.15 | 74.74 | 68.84 | 63.40 | 58.39 | 53.78 | 49.53 | 45.62 | 42.02 |
| Guizhou Province | 45.74 | 49.86 | 50.20 | 50.55 | 50.90 | 51.25 | 51.60 | 51.95 | 52.31 | 52.67 | 53.04 | 53.40 |
| Yunnan Province | 60.00 | 65.81 | 61.96 | 58.34 | 54.92 | 51.71 | 48.69 | 45.84 | 43.16 | 40.63 | 38.25 | 36.02 |
| Xizang | 5.10 | 5.30 | 5.28 | 5.25 | 5.23 | 5.20 | 5.18 | 5.16 | 5.13 | 5.11 | 5.08 | 5.06 |
| Shaanxi Province | 38.20 | 43.59 | 41.82 | 40.12 | 38.50 | 36.93 | 35.43 | 34.00 | 32.62 | 31.29 | 30.02 | 28.80 |
| Gansu Province | 32.12 | 33.27 | 31.46 | 29.75 | 28.14 | 26.61 | 25.17 | 23.80 | 22.51 | 21.29 | 20.13 | 19.04 |
| Qinghai Province | 8.62 | 9.05 | 8.55 | 8.07 | 7.63 | 7.21 | 6.81 | 6.43 | 6.07 | 5.74 | 5.42 | 5.12 |
| Ningxia | 8.59 | 9.55 | 9.32 | 9.11 | 8.89 | 8.68 | 8.48 | 8.28 | 8.09 | 7.90 | 7.71 | 7.53 |
| Xinjiang | 36.71 | 39.76 | 32.91 | 27.24 | 22.55 | 18.66 | 15.44 | 12.78 | 10.58 | 8.76 | 7.25 | 6.00 |
| amount to | 1567.95 | 1799 | 1639.15 | 1496.37 | 1368.7 | 1254.44 | 1152.05 | 1060.21 | 977.7 | 903.6 | 836.9 | 776.8 |
| National actual value | 1655 | 1786 | 1723 | 1523 | 1465 | 1200 | 1062 | 956 | 902 | | | |
| National forecast | 1655 | 1849.17 | 1672.48 | 1512.67 | 1368.13 | 1237.4 | 1119.16 | 1012.23 | 915.5 | 828 | 748.9 | 677.3 |

to 2021. There is a large gap between the results and the cumulative value of provinces and cities, which must be tested in time. In addition, according to the "China Population Forecast Report 2023 edition" released by Yuwa Population Research, the total population of China will fall to 479 million by 2100 without a substantial birth burden reduction policy. According to the three fertility parameters of high, medium, and low fertility, three prediction schemes can be formed for births from 2023 to 2050. As shown in Fig 6, none of the three predictions shows a significant increase in the birth number, which also means that China will soon face the problem of fewer children [39]. Therefore, it is necessary to take effective measures while adjusting fertility policy to deal with the grey rhinoceros of low fertility traps.

Fig 8 shows the birth population predictions under different fertility rate parameters from 2023 to 2050. The high prediction (orange line) indicates that the number of births rose slightly initially and then stabilized, eventually dropping to 9.11 million in 2050. The medium forecast (the yellow line) is relatively stable, slowly decreasing from 9 million to 6.98 million. The low forecast (blue line) shows a significant downward trend, dropping from 9 million to 4.93 million. The above three predictions all indicate that the number of future births will be affected by the fertility rate and there is a downward trend.

## Conclusion and policy implications

This study evaluated the net effect of fertility policy adjustments based on provincial panel data and found that the universal two-child and three-child policies had a limited effect on increasing the fertility rate, and there was significant regional heterogeneity. The research results are highly consistent with the theoretical predictions of Becker's fertility cost-benefit model: On the one hand, the continuous increase in the opportunity cost of raising children (such as the risk of career interruption for women) and direct costs (education and housing expenditures) leads to a diminishing marginal interest in fertility released by policies (see Fig 4). The Becker model emphasizes the dominant role of economic rationality within the family in fertility decisions. However, this study further reveals that the moderating effect of policy intervention needs to be analyzed in combination with the institutional environment - for example, in urban areas, due to the higher coverage rate of childcare services (about 30%), the policy elasticity coefficient (0.15) is significantly higher than that in rural areas (0.07) [40].

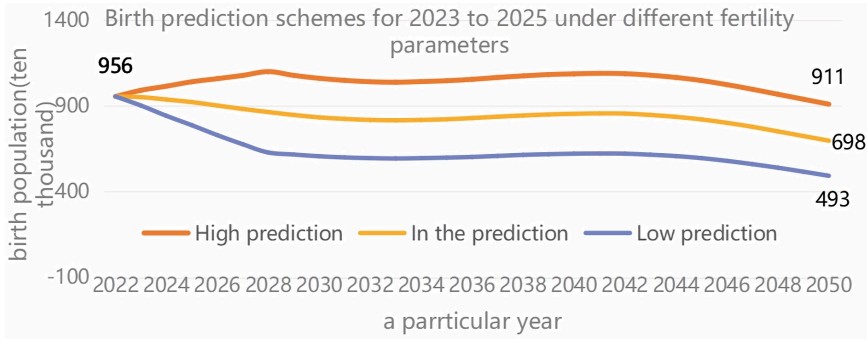

**Fig 8. 2023-2050 (ten thousand).** Note: The data are from yu Wa population study. This chart presents the birth population prediction schemes under different fertility parameters from 2023 to 2025, divided into three predicted values: high, medium, and low (for example, high prediction 9.56 million, medium prediction 6.98 million, and low prediction 4.93 million). The data in the figure shows a clear decreasing trend and includes negative values (-100), which may reflect the predicted scenarios under extreme assumptions.

It confirms the assertion that insufficient institutional support will intensify the "cost suppression effect" in the Becker model. On the other hand, the law of diminishing marginal utility of fertility is verified here: The response intensity of families to the second and third children is only 40%-60% of that to the first child, indicating that simply easing fertility restrictions is difficult to break through the utility threshold.

Based on the above findings, the following policy suggestions are put forward. Firstly, to address the challenge of insufficient effect of fertility policy adjustments, it is necessary to construct a multi-level and precise support system to counteract the cost suppression effect and reshape the fertility utility function [41]. First of all, a stepwise fertility subsidy mechanism should be established, and progressive incentive policies should be designed based on the difference in the number of children. For instance, referring to the child-rearing allowance model in Sweden, a one-time reward of 5% of the annual income is provided to families with first children [42]. The subsidy ratio for second and third children is gradually increased to 8% and 12% respectively. The subsidy period covers the critical growth period from 0 to 6 years old, alleviating the impact of diminishing marginal utility in Becker's theory. At the same time, the responsibility of fathers in raising children should be strengthened. Non-transferable paid leave exclusive to fathers (such as the "90-day father's leave" in South Korea) should be implemented, requiring the mandatory use of at least 30 days, with a salary replacement rate of no less than 70%, and the cost-sharing mechanism within the family should be restructured through institutional constraints.

Secondly, it is necessary to accelerate the improvement of public services to reduce opportunity costs. In the field of childcare, by 2025, the number of childcare places per 1,000 people will be achieved at 4.5, with a focus on rural areas. Additionally, enterprises will be offered preferential policies such as a 50% reduction in property tax for self-built childcare centers, and the rate of postpartum women returning to work will be increased by 20% to 30%. In the field of elderly care, the pilot scope of long-term care insurance will be expanded, the reimbursement ratio for the care expenses of disabled elderly people will be raised to 65%, and the "time bank" mutual assistance elderly care model will be promoted to alleviate the dual pressure on the elderly and children.

Furthermore, a fertility-friendly environment should be created through legislation and institutional reform. It is mandatory for enterprises to allow over 30% of their positions to work remotely and incorporate "child-friendliness" into the ESG rating system of listed companies, compelling them to optimize human resource management. In the Yangtze River Delta and Pearl River Delta regions, the basic education system has been piloted to be shortened by one year. Through curriculum integration and digital teaching to enhance efficiency, the window period for marriage and childbearing has been advanced to 22-25 years old, alleviating the pressure on the fertility rate caused by late marriage and late childbearing.

Finally, a dynamic policy calibration mechanism needs to be established to enhance regional adaptability [43]. Based on the "proportion of urban population - per capita GDP-sex ratio", a fertility response index is constructed. For the central and western provinces, the special transfer payment for childcare infrastructure is increased (with an average annual growth rate of $\geq 15\%$), while for the high-cost eastern regions, the "fertility points-based household registration" policy is implemented, meaning that families with multiple children can obtain priority household registration qualifications. The policy's targeting is enhanced through differentiated tools. These measures not only echo the core logic of the Becker model regarding the cost-utility trade-off, but also expand "family rationality" to "institutional-family collaborative rationality" through institutional innovation, providing a systematic solution to break the low fertility trap.

## Author contributions

**Conceptualization:** Wei Wang.

**Data curation:** Wei Wang, Yalan Mo, Yanxi Kuang.

**Formal analysis:** Yanxi Kuang.

**Funding acquisition:** Yalan Mo.

**Methodology:** Wei Wang, Yanxi Kuang.

**Writing – original draft:** Wei Wang, Yanxi Kuang.

**Writing – review & editing:** Wei Wang, Yalan Mo.

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
