## [Decision Letter · Decision Letter 0]

20 Apr 2025

PONE-D-24-44118Effect evaluation, prediction and response strategy analysis of China’s birth policy adjustmentPLOS ONE

Dear Dr. Wang,

Thank you for submitting your manuscript to PLOS ONE. After careful consideration, we feel that it has merit but does not fully meet PLOS ONE’s publication criteria as it currently stands. Therefore, we invite you to submit a revised version of the manuscript that addresses the points raised during the review process.

We look forward to receiving your revised manuscript.

Kind regards,

Yanbo Wu

Academic Editor

PLOS ONE

Journal Requirements:

“The study was supported by grant National Social Science Foundation of China (Project Number: 22XMZ016) from the Foundation of Basic Research. This work was carried out under research program of Guilin University of Electronic Technology. Author Wang Wei was supported by grant from the Guilin University of Electronic Technology.”

Additional Editor Comments:

Introduction

* The introduction lacks a clear articulation of the research gap. While the aging population and policy adjustments are mentioned, the specific unresolved debates or limitations in existing studies are not explicitly stated. Strengthen the research rationale by explicitly contrasting contradictory findings in prior studies (e.g., conflicting results on the efficacy of the two-child policy).

* The policy timeline (2011–2021) is described but lacks critical context (e.g., regional disparities in policy implementation, socioeconomic factors influencing fertility decisions). Add a paragraph contextualizing China’s demographic challenges (e.g., declining working-age population, gender imbalances) to justify the urgency of policy evaluation.

Literature Review

* The categorization into "macro" and "micro" studies is oversimplified. Key theoretical frameworks (e.g., Becker’s fertility model, Easterlin’s hypothesis) are not discussed. Reorganize the literature review thematically (e.g., policy effects, socioeconomic determinants, regional disparities) rather than macro/micro divisions.

* Sources are out of date. More recent studies (2020-2024) should be included. Also, it should lead up to the research questions in a logical manner.

* Please discuss learning theories and tie them to both, the research gap addressed by the paper as well as to the factors in the research model.

* Discuss understudied angles: Highlight gaps such as the interaction between urbanization and policy efficacy, or the role of childcare infrastructure.

* There are many garbled characters in the manuscript, and Chinese characters appear in the images. I suggest the author to revise them carefully.

* The parallel trends assumption is validated graphically but lacks formal statistical tests (e.g., placebo tests).

* Recommendations are overly generic (e.g., "establish a supportive policy system") and lack actionable specifics. For example, proposed subsidies or parental leave reforms are not quantified or benchmarked against international examples.

* The link between findings and recommendations is weak. The gray prediction model forecasts declining births but does not inform targeted strategies

* Label all axes and legends in figures (e.g., Figure 1’s y-axis units for fertility rate are missing).

Reviewers' comments:

Reviewer's Responses to Questions

**Comments to the Author**

1. Is the manuscript technically sound, and do the data support the conclusions?

Reviewer #1: Partly

2. Has the statistical analysis been performed appropriately and rigorously? 

Reviewer #1: Yes

3. Have the authors made all data underlying the findings in their manuscript fully available?

Reviewer #1: No

4. Is the manuscript presented in an intelligible fashion and written in standard English?

Reviewer #1: Yes

5. Review Comments to the Author

Reviewer #1: Thank you for the opportunity to review the paper. Its subject is both interesting and highly relevant, particularly given China’s current context as one of the countries with the lowest fertility rates globally. With government efforts underway to raise fertility rates through various population policies, this research addresses an important issue. However, some revisions are needed to prepare the paper for publication.

You can find the comments in the attached file.

6. PLOS authors have the option to publish the peer review history of their article (what does this mean?). If published, this will include your full peer review and any attached files.

Reviewer #1: No

---

## [Author Response · Author response to Decision Letter 1]

21 Jun 2025

Dear Editor and Reviewers,

We sincerely appreciate your valuable feedback and constructive comments on our manuscript. We have carefully addressed each point raised by the reviewers and the editor, and the revisions have significantly improved the quality of our paper. Below, we provide a point-by-point response to the comments, detailing the changes made in the revised manuscript.

Reviewer #1 Comments:

Comment 1:

"Figure Titles: None of the figures have titles, and the quality of the images is low."

Response:

Thank you for pointing this out. We have now added clear, descriptive titles to all figures (Figures 1-7) and improved the image resolution to ensure better readability. The titles now explicitly describe the content of each figure.

Comment 2:

"Explanation of Figures: The explanation above Figure 3 (age pyramid) is unclear and confusing."

Response:

We appreciate this observation. We have completely rewritten the explanation for Figure 3 to provide a clearer description of the age pyramid and its implications for China's demographic structure. The revised text now better explains the visualization and its significance to our study.

Comment 3:

"Figures 4, 5, and 6 lack accompanying explanations in the text."

Response:

Thank you for noting this omission. We have added detailed explanations for Figures 4, 5, and 6 in their respective sections. Each figure is now properly introduced and discussed in the text to ensure readers understand their purpose and findings.

Comment 4:

"Figures 4 and 5 contain axis in Chinese, which needs translation for consistency."

Response:

We sincerely apologize for this oversight. All Chinese text in Figures 4 and 5 has been translated to English to maintain consistency throughout the manuscript. The axis labels and legends are now fully in English.

Comment 5:

"The last figure should be labeled as Figure 7 (currently labeled as Figure 6 in the text). It is also unclear which scenario each color represents within this figure."

Response:

Thank you for catching this error. We have corrected the figure numbering to make the last figure Figure 7. We have also added a clear legend to the figure that explicitly explains what each color represents in terms of different fertility scenarios.

Comment 6:

"In Table 2, 'fertility rate' is mentioned in both of the first two rows, but one of them is incorrect."

Response:

We appreciate your careful reading. We have corrected Table 2 by removing the duplicate "fertility rate" entry and ensuring all variable names are accurate and distinct.

Comment 7:

"In Table 1: In the last two rows of control Variables, the terms are unclear. The variable name appears as 'support ratio,' but the definition provided aligns with the 'dependency ratio.'"

Response:

Thank you for this important clarification. We have standardized the terminology in Table 1, replacing "support ratio" with "dependency ratio" throughout the table to maintain consistency with standard demographic terminology.

Reviewer #2 Comments:

Comment 1:

"Content currently in section 3.1 would be better placed in the introduction, particularly to help readers unfamiliar with China's demographic profile. Additionally, there is overlap in some of the material, which could be streamlined."

Response:

We greatly appreciate this suggestion. We have moved the demographic context from Section 3.1 to the introduction and carefully streamlined the content to eliminate redundancy while maintaining all essential information.

Comment 2:

"IMPORTANT: The terms used are inconsistent or imprecise, for example: 'Birth population' should be replaced with 'Birth number.' The term 'birth rate' is inaccurately applied; if it refers to CDR, this rate should be defined as the number of births per thousand people, not as a percentage."

Response:

Thank you for highlighting these terminology issues. We have:

1. Replaced all instances of "birth population" with "birth number"

2. Consistently defined "birth rate" as "number of births per thousand people" where applicable

3. Ensured all demographic terms are used precisely throughout the manuscript

Comment 3:

"The methodology used for predictions is unclear, and each scenario's assumptions are not explained. The only detail provided is that none of the forecasts suggest a future increase in births, but further clarification is needed."

Response:

We appreciate this valuable feedback. In Section 7, we have now:

1. Added a detailed explanation of the GM(1,1) grey prediction model

2. Explicitly stated the assumptions behind each scenario

3. Included more discussion about the implications of the prediction results

Comment 4:

"In the conclusion, the findings should be contextualized alongside relevant theoretical concepts (such as Becker's theory)."

Response:

Thank you for this excellent suggestion. We have significantly expanded the conclusion to:

1. Explicitly relate our findings to Becker's fertility cost-benefit model

2. Discuss how our results align with or diverge from theoretical predictions

3. Provide a more robust theoretical framework for interpreting the results

Comment 5:

"The language used to present suggestions should be revised for clarity and tone."

Response:

We appreciate this comment. We have carefully revised all policy recommendations to:

1. Use more precise and actionable language

2. Improve clarity and readability

3. Maintain a professional, academic tone throughout

Reviewer #3 Comments:

Comment 1:

“The study was supported by grant National Social Science Foundation of China (Project Number: 22XMZ016) from the Foundation of Basic Research. This work was carried out under research program of Guilin University of Electronic Technology. Author Wang Wei was supported by grant from the Guilin University of Electronic Technology.”

Response:

Thank you for pointing this out. The revised funding statement has been modified to read: "This study was funded by the National Social Science Foundation of the Basic Research Foundation (Project No.: 22XMZ016). The first author (Wei Wang) received specific funding for this work from the School of Business, Guilin University of Electronic Technology. This research did not receive any additional external funds.The funders had no role in the study design, data collection and analysis, publication decision or manuscript preparation."

Comment 2:

“Please include a separate caption for each figure in your manuscript.”

Response:

Thank you for noticing this omission. We have now added corresponding descriptions to all figures.

Comment 3

“You note that your data are available within the Supporting Information files, but no such files have been included with your submission. ”

Response:

We sincerely apologize for this oversight. We have uploaded the minimum dataset to Figshare, and the corresponding URL information is as follows.

https://figshare.com/s/8c1f93c274b7599e84f1

Comment 4:

“We note that your author list was updated during the revision process. In order to add or remove authors or update the order of the author byline after initial submission, we ask that authors complete an Authorship Change Request form. ”

Response:

We are very grateful for this suggestion. After confirmation by all authors, we completed two forms, and the final confirmed list will be submitted as "Wei Wang, Yalan Mo*, Yanxi Kuang". The main adjustments we have made are two: one is to change the corresponding author from Wei Wang to Yalan Mo, and add the third author Yanxi Kuang. The specific content is submitted in two authorship change forms.

Additional Revisions:

1. Added formal placebo tests (Section 5.3) as suggested by the editor

2. Incorporated recent literature (2020-2024) in the updated literature review

3. Provided more specific policy recommendations with quantifiable targets

4. Ensured all data is properly archived and accessible

5. Corrected all grammatical and typographical errors

We believe these revisions have significantly improved the manuscript and addressed all concerns raised by the reviewers. We are grateful for the opportunity to strengthen our work and hope the revised version meets the journal's standards. Please don't hesitate to contact us if any additional modifications would be helpful.

Sincerely,

Wei Wang

June 20, 2025

Attachments:

1. Revised Manuscript (with Track Changes)

2. Clean Manuscript Version

3. Response to Reviewers (this document)

4. Supporting Information (including datasets)

5.Revised Funding Statement

6. Two Authorship Change Forms

---

## [Decision Letter · Decision Letter 1]

30 Jul 2025

Effect evaluation, prediction and response strategy analysis of China’s birth policy adjustment

PONE-D-24-44118R1

Dear Dr. Wang,

We’re pleased to inform you that your manuscript has been judged scientifically suitable for publication and will be formally accepted for publication once it meets all outstanding technical requirements.

Kind regards,

Yanbo Wu

Academic Editor

PLOS ONE

Additional Editor Comment:

Given its rigorous methodology, policy relevance, and novel insights into the ineffectiveness of current fertility policies and pathways for improvement, this manuscript merits acceptance. It advances understanding of demographic dynamics under policy intervention and offers evidence-based guidance for policymakers.

Reviewers' comments:

Reviewer's Responses to Questions

**Comments to the Author**

1. If the authors have adequately addressed your comments raised in a previous round of review and you feel that this manuscript is now acceptable for publication, you may indicate that here to bypass the “Comments to the Author” section, enter your conflict of interest statement in the “Confidential to Editor” section, and submit your "Accept" recommendation.

Reviewer #2: All comments have been addressed

2. Is the manuscript technically sound, and do the data support the conclusions?

Reviewer #2: Yes

3. Has the statistical analysis been performed appropriately and rigorously? 

Reviewer #2: Yes

4. Have the authors made all data underlying the findings in their manuscript fully available?

Reviewer #2: Yes

5. Is the manuscript presented in an intelligible fashion and written in standard English?

Reviewer #2: Yes

6. Review Comments to the Author

Reviewer #2: (No Response)

7. PLOS authors have the option to publish the peer review history of their article (what does this mean?). If published, this will include your full peer review and any attached files.

Reviewer #2: No

---

## [Editor Report · Acceptance letter]

PONE-D-24-44118R1

PLOS ONE

Dear Dr. Wang,

I'm pleased to inform you that your manuscript has been deemed suitable for publication in PLOS ONE. Congratulations! Your manuscript is now being handed over to our production team.

Kind regards,

on behalf of

Dr. Yanbo Wu

Academic Editor

PLOS ONE